# Location Method of Buried Polyethylene Gas Pipeline Based on Acoustic Signal Ellipse Method

**DOI:** 10.3390/s24227302

**Published:** 2024-11-15

**Authors:** Hanyu Zhang, Yang Li, Zhuo Xu, Ao Zhang, Xianfa Liu, Pengyao Sun, Xianchao Sun

**Affiliations:** 1School of Mechanical Engineering, Northeast Electric Power University, Jilin 132011, China; 13760944553@163.com (H.Z.); liyang891209@126.com (Y.L.); 2School of Mechanical Engineering, Shenyang Jianzhu University, Shenyang 110168, China; zhangao2902@sjzu.edu.cn; 3Hunan Anguang Inspection and Testing Company, Changde 415131, China; 19974019370@163.com; 4Special Equipment Inspection Center of Jilin, Special Equipment Accident Investigation Service Center of Jilin, Jilin 132089, China; sageroc@163.com; 5Changchun Special Equipment Inspection & Research Institute (Changchun Special Equipment Safety Monitoring Center), Changchun 130013, China; sunxc@cctj.freeqiye.com

**Keywords:** elliptic equation localization, cross-correlation delay time, COMSOL simulation depth positioning, buried PE pipeline

## Abstract

This study proposes a buried PE gas pipeline positioning method based on the elliptical method of an acoustic signal analysis. The cross-correlation time delay positioning technology is combined with the elliptical equation, forming an effective mechanism for pipeline depth positioning. First, a dual-tree complex wavelet transform is employed to denoise the collected signals, enhancing the quality and accuracy of the data. Subsequently, the cross-correlation function is utilized to extract the delay times between the signals. The obtained delay times are then substituted into the elliptical equation to calculate the depth of the buried PE pipeline. Based on this theoretical framework, a simulation model is established in COMSOL, and positioning simulation analyses are conducted under three different conditions: pipeline depth, relative sensor positions, and distances between sensors and excitation points. The simulation results indicate that a clear correlation exists between the signal delay time and the pipeline position, with simulation errors controlled within 5%, thus validating the theoretical feasibility of the method. To further assess the effectiveness of this approach, an experimental testing system is constructed. The experimental study was carried out under four different conditions: pipeline burial depth, relative sensor positions, distances between sensors and excitation points, and excitation frequencies. The experimental results demonstrate that these factors significantly affect the pipeline depth positioning. The comparison results show that the method has a high accuracy in depth positioning, with experimental errors controlled within 10%. This study proves that accurate positioning of pipeline depth could be achieved by substituting signal delay times into the elliptical equation, thereby validating the method’s feasibility in practical applications. The proposed method effectively addressed the shortcomings of existing pipeline depth positioning technologies, providing important theoretical support and a practical reference for future pipeline positioning research.

## 1. Introduction

A buried pipeline system is a crucial component of urban infrastructure [1,2,3]. Due to the advantages of non-metallic pipelines, such as a low cost and corrosion resistance, they have found significant applications in the construction of buried pipeline networks. Among these, polyethylene (PE) pipelines are the most widely used in urban gas pipeline construction [4,5]. However, as these pipelines are subjected to soil environments for extended periods, the buried pipeline systems are highly susceptible to aging, cracking, and other forms of damage [6,7,8]. Additionally, for some buried pipelines that have been in place for a long time, it is challenging to accurately determine their installation routes and positions. Therefore, there is an urgent need to develop an efficient method for precisely locating buried PE pipelines, to ensure the safety of urban underground networks.

Currently, the primary methods for locating buried pipelines involve ground-penetrating radar (GPR) and electromagnetic wave technology. Srinivas et al. [9] conducted a detection analysis of buried pipelines using GPR, and validated the method through imagery, facilitating real-time assessment and satisfactory “spatial diversity”. Kavi et al. [10] combined GPR-based nondestructive testing (NDT) techniques with innovative strategies, which is vital for the next generation of pipeline applications. Luo et al. [11] employed information acquisition, image processing techniques, damage prediction models, and pipeline diagnosis systems for the intelligent perception and precise identification of urban underground drainage networks, outlining future research directions for intelligent pipeline diagnoses. López et al. [12] summarized the methods for ground-penetrating radar (GPR) positioning and proposed improvements to the positioning system that incorporated synthetic aperture radar (SAR) technology into GPR. This integration resulted in the development of a GPR-SAR system capable of producing high-resolution microwave images. Rudolph et al. [13] proposed the application of lightweight electromagnetic induction (EMI) sensors to assess the spatial variability of soil. They determined management zones by mapping the soil’s apparent electrical conductivity. Mat Junoh et al. [14] conducted tests on the integration of the geophysical principles between electromagnetic locators (EMLs) and ground-penetrating radar (GPR). The results indicated that both EML and GPR were effective methods for detecting pipeline diameters. Also, they emphasized the necessity of field validation and the appropriate selection of antenna frequencies. Hoarau et al. [15] proposed a method that could capture the signal of interest, reduce noise, and provide local estimates of relative permittivity, effectively detecting pipelines with low response levels while maintaining a reasonable probability of a false alarm (PFA). Ambruš et al. [16] evaluated a robust concept through simulations and experiments, demonstrating effective pipeline shape estimation despite uncertainties in sensor positioning and pipeline geometry using limited spatial diversity.

Many scholars have analyzed acoustic signals generated by impacts [17] or other excitation methods to locate buried pipelines. Cui et al. [18] studied leak location technology based on low-frequency narrowband acoustic emission, which controlled the leak location error within 5%. Xu et al. [19] experimented with a multi-level frame leak location method in a buried pipeline with continuous leakage sources. When the sensor spacing was 10–33 m, the maximum positioning error was 5.3%. Lang et al. [20] proposed a fast and effective method to locate small leaks by using the information fusion method, combining ultrasonic sound velocity signals with flow signals. Xiao et al. [21,22] first proposed a comprehensive acoustic signal leakage detection method by using a wavelet transform and support vector machine (SVM), and established a correlation function model of gas pipeline leakage noise, providing a theoretical and experimental basis for optimizing gas pipeline leakage detection and location. Li et al. [23] studied the influence of environmental noise, the welding seam, anti-corrosion coating, and other factors on field measurement in a specific application scenario, and located the leakage point by using a discrete wavelet transform and time-spectrum method. Zheng et al. [24] identified gas pipeline leakage points through leakage noise in soil, and the positioning error in the experimental scene was 8%~12%. Yan et al. [25] investigated a method for mapping and locating underground pipeline leaks using imaging technology. By employing a group of acoustic vibration sensors to measure surface vibrations, their results aligned closely with the actual leak locations. Chen et al. [26] proposed an effective and validated non-isothermal model to investigate the mechanisms of non-permanent wave (NPW) generation and propagation in long-distance gas pipelines. The proposed leak signal characterization and preprocessing techniques reliably identified and accurately located actual leakage events within 36 s. Zuo et al. [27] developed a gas pipeline leakage monitoring algorithm utilizing a distributed acoustic sensing (DAS) system. This algorithm could capture the time-domain signal characteristics of pipeline leaks, enabling leak identification and the spatial localization of the leak points in the frequency domain. Zhang et al. [28] investigated the time–frequency signals of acoustic waves generated by pipeline leaks using a “sound-pipe, sound-pressure” multiphysics coupling approach. The proposed method enhanced the detection capability for small leaks and provided a novel pathway for the promotion and engineering application of pipeline leakage detection technology. Ndalila et al. [29] studied the significant characteristics of dynamic pressure fluctuations in gas using computational fluid dynamics (CFD) modeling methods. They conducted transient simulations of the model, revealing the impact of one and two leak points on the dynamic pressure within the pipeline. Li et al. [30] combined the principles of fluid dynamics with Lighthill’s acoustic analogy theory to theoretically model the propagation of leakage noise in water supply pipelines. The findings provided theoretical guidance and support for the analysis of acoustic signal characteristics associated with pipeline leaks and the identification of leakage conditions. Zhang et al. [8] proposed a novel approach for identifying leaks in buried natural gas pipelines by analyzing leakage noise in the soil. Experimental results indicated that this method achieved a localization error of 8% to 12% for leaks in buried gas pipelines. Ahmad et al. [31] proposed a reliable algorithm for pipeline leak detection using acoustic emission signals. The algorithm achieved high classification accuracy in detecting leaks under various leak sizes and fluid pressures. Chen et al. [5] evaluated the use of distributed acoustic sensing (DAS) for detecting pinhole gas leaks in buried pipelines within sandy soils. The gas–fiber friction exhibited a broader spectral response; however, it reduced the peak frequency and amplitude generated by soil strain. These findings provided a basis for improving the monitoring of pinhole leaks in buried gas pipelines using DAS technology. In contrast, only a limited number of researchers have focused on the localization of acoustic signals from non-leaking buried pipelines. Zhang et al. [32] proposed a monitoring method that integrated multiple signal sources from both inside and outside the pipeline. This approach significantly enhanced detection sensitivity, localization accuracy, and response speed. Witos et al. [33] introduced a novel method for the detection of leaks in metallic pipelines. The study presented research findings obtained from both laboratory experiments and practical applications. Acoustic emission sensors were installed to measure the attenuation curves of two measurement paths associated with the sensors. Subsequently, measurements were conducted according to the proposed methodology. Finally, the recorded signals were analyzed to determine the location of the leak source. However, most of the methods discussed in the above studies are mainly used to locate the leakage points of buried pipelines. Only a few researchers focus on the field of positioning methods for pipelines. Hei et al. [34] proposed a novel method for calculating time of arrival, termed TOAIP (Time of Arrival with Instant Phase), which eliminated the need for manual threshold selection. This method determined the time of arrival (TOA) using the first zero-crossing of the signal and the instantaneous phase derived from the periodicity of the phase. Additionally, a two-dimensional impact localization (TDIL) technique was developed to simultaneously estimate impact position information along both length and circumferential dimensions. Huang et al. [35] proposed a new method for locating impact sources based on the time-reversed virtual focusing triangulation technique. This approach selected key sensors through energy power filtering, extracted narrowband Lamb wave signals using wavelet packet decomposition, and performed synthesis. Experimental results indicated that under non-motorized conditions, the average error of this method was 0.89 m, while under motorized conditions, the average error was 1.12 m.

In summary, current positioning technologies for buried pipelines primarily rely on ground-penetrating radar and electromagnetic wave techniques. However, these methods exhibit significant positioning errors in practical applications and have stringent environmental requirements. To address this issue, this study proposes a buried polyethylene (PE) pipeline positioning method based on the elliptical method of an acoustic signal analysis. This approach integrates dual-tree complex wavelet denoising technology, cross-correlation time delay positioning technology, and the elliptical positioning method. Due to the environmental impact of buried pipelines, there will be a lot of background noise in the collected acoustic signals. To obtain more accurate pipeline location information, the collected acoustic signals should be denoised. The dual-tree complex wavelet is an effective denoising method, which has been applied in some references mentioned above. In this study, the dual-tree complex wavelet is used to denoise the collected acoustic signals to reduce the impact of noise on localization accuracy. After denoising, cross-correlation processing is employed to accurately calculate the delay time between two signals. When the delay time τ = 0, it indicates that the intermediate sensor is located directly above the buried pipeline. Subsequently, an elliptical equation was established, and the delay times of the remaining two signals were substituted into it to achieve the vertical depth positioning of the buried pipeline. Based on this positioning theory, a simulation model was developed to validate the feasibility of the proposed method. Furthermore, an experimental testing system was constructed to conduct a series of positioning experiments under varying conditions of pipeline burial depth, sensor positions, excitation point locations, and excitation frequencies, thereby verifying the reliability of the proposed method. The buried PE pipeline positioning method introduced in this study not only addresses the limitations of existing methods but also provides significant theoretical guidance for the accurate localization of buried PE pipelines in future research.

## 2. Theory

As shown in Figure 1, the elliptical method for pipeline positioning consists of a model in which three sensors are arranged equidistant above the pipeline. Initially, the signals collected by the sensors on the left and right sides are subjected to denoising and cross-correlation processing to obtain the time delay between the two signals. When the delay time $\tau_{0}\;$ = 0, it indicates that the middle sensor is positioned directly above the buried pipeline, thus determining the lateral position of the pipeline. An elliptical equation is then established using the middle sensor and any one of the adjacent sensors as the focus. This equation can be utilized to determine the vertical depth of the buried pipeline.

From reference [36], the dual-tree complex wavelet transform has been proven to be effective for denoising leak acoustic signals in pipelines. In this study, to denoise the collected acoustic signals, the dual-tree complex wavelet theory is adopted for depth localization in buried pipelines. First, the signal *x*_1_(*x*_1_,*t*) from sensor 1 and the signal *x*_2_(*x*_2_,*t*) from sensor 2 is denoised using this technique. Then, the time delay $\tau_{0}$ is estimated by the peak of the cross-correlation function between the two signals, where the cross-correlation function $R_{{\tilde{x}}_{1}{\tilde{x}}_{2}}(\tau_{0})$ can be expressed as follows [22]:(1)$$R_{{\tilde{x}}_{1}{\tilde{x}}_{2}}(\tau_{0})=E\left[x_{1}(x_{1},t)x_{2}(x_{2},t+\tau )\right]$$
where *E*[ ] denotes the expectation operator.

The cross-spectral density $S_{{\tilde{x}}_{1}{\tilde{x}}_{2}}(\omega )$ between signals $x_{1}(x_{1},t)$ and $x_{2}(x_{2},t)$ can be expressed in the following form:(2)$$\begin{array}{ll} S_{{\tilde{x}}_{1}{\tilde{x}}_{2}}(\omega ) & =\frac{1}{2\pi }\underset{T\to \infty }{\lim}E\left[\frac{s_{1T}^{*}(x_{1},\omega )s_{1T}(x_{2},\omega )}{T}\right]\\ & =S_{0}\left(\omega \right)\psi \left(\omega \right)e^{i\omega \tau_{0}} \end{array}$$
where $\Psi \left(\omega \right)=\left|H_{1}^{*}\left(\omega ,d_{1}\right)H_{2}\left(\omega ,d_{2}\right)\right|=e^{-\alpha \sqrt{\omega }\left(d_{1}+d_{2}\right)}$, $S_{0}\left(\omega \right)=1/2\pi \left(\underset{T\to \infty }{\lim}E\left[s_{0}^{*}\left(\omega \right)s_{0}\left(\omega \right)/T\right]\right)$ is the noise spectrum at *x* = 0, and $\tau_{0}=-\left(d_{2}-d_{1}\right)/c$ is the time delay.

The noise spectrum *S*_0_(ω) is expressed in physical form to predict the correlation function. And the measured cross-spectral density can be expressed as follows [37]:(3)$$S_{{\tilde{x}}_{1}{\tilde{x}}_{2}}(\omega )=\frac{8\rho_{f}^{2}c^{2}{\bar{u}}^{2}}{\pi^{4}}{\left(\frac{a}{R}\right)}^{4}\frac{\Lambda }{U}\frac{e^{-a\sqrt{\omega }(d_{1}+d_{2})}}{1+{\left(\frac{\omega \Lambda }{U}\right)}^{2}}e^{i\omega \tau_{0}}$$

The multiplication in the frequency domain corresponds to convolution in the time domain; therefore, the cross-correlation function can be expressed by the following equation:(4)$$R_{{\tilde{x}}_{1}{\tilde{x}}_{2}}(\tau )=\frac{8\rho_{f}^{2}c^{2}{\bar{u}}^{2}}{\pi^{4}}{\left(\frac{a}{R}\right)}^{4}\frac{\Lambda }{U}F^{-1}\left\{\frac{e^{-a\sqrt{\omega }(d_{1}+d_{2})}}{1+{\left(\frac{\omega \Lambda }{U}\right)}^{2}}\right\}⊗\delta \left(\tau +\tau_{0}\right)$$
where $F^{-1}\{\;\}$ denotes the inverse Fourier transform, and ⊗ represents the convolution operator.

Expressing Equation (4) in dimensionless form, the cross-correlation function can be represented as
(5)$$R_{{\tilde{x}}_{1}{\tilde{x}}_{2}}(\tilde{\tau })=\frac{16\rho_{f}^{2}c^{2}{\bar{u}}^{2}}{\pi^{4}}{\left(\frac{a}{R}\right)}^{4}\int_{0}^{\infty }\frac{e^{-\sqrt{\Omega }({\tilde{d}}_{1}+{\tilde{d}}_{2})}e^{i\sqrt{\Omega }({\tilde{\tau }}_{0}+\tilde{\tau })}}{1+\Omega^{2}}d\Omega$$
where Ω=ωΛ/U is the dimensionless frequency, $\tilde{\tau }=\tau U/\Lambda$ is the dimensionless time lag, ${\tilde{\tau }}_{0}=\tau_{0}U/\Lambda$ is the dimensionless time delay, and ${\tilde{d}}_{i}$ is the dimensionless distance. ${\tilde{d}}_{i}$ can be expressed as
(6)$${\tilde{d}}_{i}=d_{i}\alpha \sqrt{\frac{U}{\Lambda }}$$

By employing a similar method, the cross-correlation function of $x_{1}(x,t)$ and $x_{2}(x,t)$ can be obtained, thereby deriving the cross-correlation coefficient:(7)$$\rho_{{\tilde{x}}_{1}{\tilde{x}}_{2}}(\tilde{\tau })=\frac{\int_{0}^{\infty }\frac{e^{-\sqrt{\Omega }({\tilde{d}}_{1}+{\tilde{d}}_{2})}e^{i\sqrt{\Omega }(\tilde{\tau }+{\tilde{\tau }}_{0})}}{1+\Omega^{2}}d\Omega }{\sqrt{\int_{0}^{\infty }\frac{e^{-2\sqrt{\Omega }{\tilde{d}}_{1}}}{1+\Omega^{2}}d\Omega ∙\int_{0}^{\infty }\frac{e^{-2\sqrt{\Omega }{\tilde{d}}_{2}}}{1+\Omega^{2}}d\Omega }}$$

When $\tilde{\tau }\;$ = −${\tilde{\tau }}_{0}$, the value of the cross-correlation coefficient is at its peak, with the peak expressed as follows [22]:(8)$$\rho_{{\tilde{x}}_{1}{\tilde{x}}_{2}}(-{\tilde{\tau }}_{0})=\frac{\int_{0}^{\infty }\frac{e^{-\sqrt{\Omega }({\tilde{d}}_{1}+{\tilde{d}}_{2})}}{1+\Omega^{2}}d\Omega }{\sqrt{\int_{0}^{\infty }\frac{e^{-2\sqrt{\Omega }{\tilde{d}}_{1}}}{1+\Omega^{2}}d\Omega ∙\int_{0}^{\infty }\frac{e^{-2\sqrt{\Omega }{\tilde{d}}_{2}}}{1+\Omega^{2}}d\Omega }}$$

The coefficient of the cross-correlation function of the two signals is calculated, and the delay time $\tau_{0}\;$ is obtained by observing the peak value of the cross-correlation number.

The elliptic equation is established with sensor 2 and sensor 3 as the focus. Because the locus center is not at the origin, the locus equation can be transformed by the standard elliptic equation given by the formula
(9)$$y=-\sqrt{b^{2}-\frac{b^{2}{\left(x-d_{2}/2\right)}^{2}}{a^{2}}}$$
where *a* is the major semi-axis of the elliptic equation, *b* is the short semi-axis of the elliptic equation, and *d*_2_ is the distance between sensors 2 and 3. The expressions of a and b are as follows:(10)$$\begin{array}{c} a=\frac{l}{2}\\ b=\sqrt{{\left(\frac{l}{2}\right)}^{2}-{\left(\frac{d_{2}}{2}\right)}^{2}} \end{array}$$

Based on the positional relationship between the pipeline and sensors 2 and 3, the relationship between *l*, *L*_1_, and *L*_2_ is derived.
(11)$$l=L_{1}+L_{2}=L_{1}+\sqrt{{L_{1}}^{2}+{d_{2}}^{2}}$$

The delay time $\tau_{0}\;$ between sensor 2 and 3 signals can be expressed as
(12)$$\tau_{0}=\frac{L_{2}-L_{1}}{\nu_{P}}$$
where *v*_p_ is the p-wave velocity, *v*_p_ = 300 m/s.

According to Equations (11) and (12), *l* is derived:(13)$$l=\frac{{d_{2}}^{2}}{\tau_{0}\nu_{p}}$$

Finally, the elliptic equation is obtained by substituting Equation (13) into Equation (9) and Equation (10); *y* is the depth of the buried pipeline:(14)$$y=-\sqrt{\left[{\left({d_{2}}^{2}/2\tau_{0}v_{p}\right)}^{2}-{\left(d_{2}/2\right)}^{2}\right]-\frac{\left[{\left({d_{2}}^{2}/2\tau_{0}v_{p}\right)}^{2}-{\left(d_{2}/2\right)}^{2}\right]{\left(x-d_{2}/2\right)}^{2}}{{\left({d_{2}}^{2}/2\tau_{0}v_{p}\right)}^{2}}}$$

This method is a combination of cross-correlation delay positioning and elliptic equation positioning. The transverse position of the buried pipeline can be obtained according to the delay time, and the depth of the buried pipeline can be obtained by substituting the delay time into the above elliptic equation.

## 3. Simulation Analysis

In this study, the proposed method for determining the lateral position and vertical depth of the pipeline based on delay time *τ* is validated using COMSOL Multiphysics 6.1. First, a model of a polyethylene (PE) pipeline with dimensions *L* = 1.5 m, *r* = 0.045 m, and *h* = 0.085 m is established in COMSOL. Sensors 1, 2, and 3 are positioned above the pipeline on the soil surface, with sensor 3 located at the midpoint between sensors 1 and 2. The burial depths of the pipeline are set at 10 cm, 20 cm, and 30 cm, as illustrated in Figure 2. The material parameters for the pipeline and soil are provided in Table 1. The axial position of the pipeline is determined by examining whether there is a delay time $\tau_{1}$ corresponding to the peak values of the time-domain signals received by sensors 1 and 2. By adjusting the positions of sensors 1 and 2, when the delay time $\tau_{1\;}$ is zero, it can be inferred that the pipeline is located between the two sensors. Subsequently, the vertical depth of the pipeline is determined by constructing an elliptical equation based on the previously discussed theory and calculating the depth of the pipeline using the delay time $\tau_{2}$ between sensors 2 and 3.

There are three variables in this simulation, which are the buried depth of the pipeline, the relative position of sensors 2 and 3, and the location of the sensor distance excitation point. The simulation is divided into three groups based on the buried depth of the pipeline. Table 2 shows the specific scheme of simulation analysis grouping.

The pipeline is excited using a Rayleigh wave, and the first step is to determine its lateral position. As shown in Figure 3, for a burial depth of 10 cm, the peak times of the signals collected by sensors 1 and 2 during the first experimental set are presented. When the time difference between sensors 1 and 2, denoted as the delay time $\tau_{1}$, is equal to zero, it confirms that sensor 3 is positioned directly above the pipeline. Subsequently, the excitation signals from sensors 2 and 3 are extracted to obtain their delay time $\tau_{2}$. As illustrated in Figure 4, for a burial depth of 10 cm, the positions of sensors 2 and 3 are 30 cm apart, and the distance from the excitation point to the sensors is 1 m, from which the peak times of the two signals are recorded.

As illustrated in Figure 4, the signals extracted from sensors 2 and 3 exhibit a delay time $\tau_{2}$, which is calculated to be 0.0016 s. Using Equation (14), the depth of the pipeline is determined to be 10.24 cm, resulting in an error of 2.4% compared to the actual burial depth. Subsequently, simulations are conducted for burial depths of 20 cm and 30 cm, with sensor distances from the excitation point set at 0.5 m, 1 m, and 1.5 m, and the sensor arrangements located 10 cm, 20 cm, and 30 cm from the center position. After obtaining the delay times for the buried polyethylene pipeline, the depths of the pipeline are calculated using Equation (14). The discrepancies between the calculated burial depths of the PE pipeline and the theoretical values are summarized in Table 3, Table 4 and Table 5.

First, with the positions of the sensors and the excitation point fixed, we discuss the localization accuracy of pipelines at different burial depths. The simulation results are presented in Table 3.

Next, with the pipeline burial depth and the position of the excitation point fixed, we discuss the localization accuracy of buried pipelines under different sensor arrangements. The simulation results are presented in Table 4.

Finally, with the sensor positions and sensor arrangements fixed, we discuss the localization accuracy of buried pipelines under different excitation point positions. The simulation results are presented in Table 5.

The simulation results indicate that the positioning accuracy of the buried polyethylene pipeline is influenced by the burial depth of the pipeline, the locations of the sensors, and the distance from the excitation point to the sensors. Specifically, when the pipeline is buried at a shallow depth, positioning accuracy improves when the distance from the sensor to the midpoint of the pipeline exceeds 10 cm, and when the distance from the excitation point to the sensors is greater than 0.5 m but less than 1.5 m. Conversely, positioning accuracy decreases under other conditions. Furthermore, the effect of burial depth on positioning accuracy is significantly greater than that of the sensor locations and the distance from the excitation point to the sensors.

## 4. Experimental Verification

Based on the theoretical and simulation foundations discussed earlier, a laboratory experiment was conducted on the pipeline. As shown in Figure 5, the experimental setup includes an MI-7004 signal acquisition system, an SALC05KE modal hammer (Yiheng Technology Co., LTD. Hangzhou, China), three PCB353B15 accelerometers (PCB Piezotronics, Inc. Buffalo, NY, USA), a movable workbench, a conduction plate, a 100 N shaker, a signal generator, and a power amplifier. A polyethylene pipeline with a length of 3.2 m is embedded in a rectangular box measuring 3 m in length, 1 m in width, and 1 m in height, at burial depths of 10 cm, 20 cm, and 30 cm. The 100 N shaker is positioned on the side of the extending PE pipeline to provide excitation, while three sensors are arranged directly above the pipeline via the conduction plate to capture the excitation acoustic waves. The software testing environment is configured as follows: (I) a frequency range of 0–5000 Hz, (II) Hamming window applied, and (III) frequency resolution set to 1 Hz.

Before the pipeline depth positioning experiment, the SALC05KE modal force hammer is first used to collect the natural frequencies of the pipeline, which are 110 Hz, 700 Hz, and 950 Hz, respectively.

### Positioning of Buried Pipelines Under Different Circumstances

Positioning experiments on the buried pipeline were conducted under four different conditions: varying burial depth, different sensor locations, different relative excitation positions, and different excitation frequencies. Initially, the pipeline is embedded in sandy soil at depths of 10 cm, 20 cm, and 30 cm, and the shaker is set to an excitation frequency of 700 Hz. Sensors are arranged at a distance of 1 m from the excitation point, with a relative distance of 30 cm between sensors 1 and 2 and sensor 3. The collected raw signals are processed using the dual-tree complex wavelet to reduce the impact of noise on localization accuracy [36]. As shown in Figure 6, the waveform diagrams of the raw and denoised signals for signal 1 and signal 3 are presented. The vibration signals of the buried pipeline are collected and analyzed to determine the respective delay times, as shown in Figure 7, Figure 8 and Figure 9. The obtained delay times are subsequently substituted into Equation (14) to calculate the pipeline depths *y*_1_ and *y*_2_, along with their associated errors, as presented in Table 6.

To analyze the effect of sensor location, the pipeline is embedded in sandy soil at a depth of 10 cm, with the relative positions of sensors 2 and 3 set at 10 cm, 20 cm, and 30 cm, respectively. The shaker is adjusted to an excitation frequency of 700 Hz, while the sensors are positioned 1 m away from the excitation point. The vibration signals of the pipeline are collected and analyzed, and the delay time of the collected signals is determined. The positions of the pipelines are then calculated using Equation (14), and the resulting errors are obtained, as shown in Table 7.

To analyze the effect of relative excitation position, the pipeline is embedded in sandy soil at a depth of 10 cm, and the shaker is set to an excitation frequency of 700 Hz. The sensors are arranged at distances of 0.5 m, 1 m, and 1.5 m from the excitation point, with a relative distance of 30 cm between sensors 2 and 3. After collecting and analyzing the vibration signals from the pipeline, the delay time of the collected signals is determined. The positions of the pipelines are then calculated using Equation (14), and the resulting errors are presented, as shown in Table 8.

To analyze the effect of excitation frequency, the pipeline is embedded in sandy soil at a depth of 10 cm, and the shaker is set to excitation frequencies of 110 Hz, 700 Hz, and 950 Hz, respectively. The sensors are positioned directly above the excitation point, with a relative distance of 30 cm between sensors 2 and 3. After excitation, the vibration signals of sensors 2 and 3 are collected and analyzed from the pipeline, and the delay time of the collected signals can be determined. The positions of the pipelines are subsequently calculated using Equation (14), and the resulting errors are presented, as shown in Table 9.

By analyzing the experimental results, the depths of the pipeline calculated from the signals collected by sensors 1 and 3 are cross-validated with the depths calculated from the signals collected by sensors 2 and 3. In the experimental environment of this study, if other conditions are constant, higher positioning accuracy can be achieved when the relative distance between the sensors exceeds 10 cm. Notably, when the relative distance reaches 30 cm, the positioning error is minimized to just 4.4%. When the pipeline is buried at a depth of 10 cm, the minimum positioning error is observed at 4.5%. The positioning accuracy is optimal when the distance of the sensors from the excitation point is within the range of 0.5 m to 1.5 m, with a minimum positioning error of 5.3%. Furthermore, the best positioning accuracy is achieved at an excitation frequency of 700 Hz, resulting in a positioning error of 5.8%. Due to limitations in the experimental equipment, this study is only able to conduct positioning experiments within a relatively small range. The analysis of experimental errors suggests that the elliptical positioning method proposed in this study can accurately determine the depth of the pipeline, with positioning errors maintained within 10%, thereby providing a feasible solution for the depth determination of buried PE pipelines.

## 5. Conclusions

This study proposes a method for locating buried PE gas pipelines based on an elliptical approach utilizing acoustic wave signals. The proposed method integrates the cross-correlation delay positioning technique with the elliptical equation. To validate the feasibility of the proposed method, simulations were conducted in COMSOL under three different situations: varying pipeline burial depths, relative positions between sensors, and distances of sensors from the excitation point. The relationship between the time difference in the two signals and the pipeline location was then determined. Subsequently, an experimental testing system was established, and experiments were conducted under four different conditions: the pipeline burial depth, relative position of sensors, distance between the sensors and the excitation point, and excitation frequency. The specific conclusions are as follows:(1)The simulation results show that under the specified simulation conditions, the burial depth of the pipeline has a significantly greater impact on localization accuracy than the relative position of the sensors and the distance between the sensors and the excitation point. This indicates that as the pipeline burial depth increases, the localization accuracy is more greatly affected. Moreover, for the cases in our simulations, the localization errors are all within 5%, demonstrating that the method has high localization accuracy.(2)The experimental results show that the relative position of the sensors, the distance between the sensors and the excitation point, excitation frequency, and the depth of the pipeline all have certain effects on the depth localization of the buried pipeline. Based on the comparison results, for different experimental conditions in our study, the localization errors are all within 10%. This indicates that the proposed localization method can be effectively used for pipeline depth localization, providing strong support for future practical applications.

Existing methods for buried pipeline localization primarily suffer from issues such as inaccurate localization, low efficiency, and complexity. The ellipse localization method proposed in this paper effectively addresses these problems. Through the combination of theory, simulation, and experiments, the validity of the method for the depth localization of buried pipelines is confirmed. Additionally, this method is characterized by its simplicity of operation and relatively low cost, providing significant engineering guidance for the localization of buried PE pipelines. This study provides important theoretical support for the further optimization of sensor placement and the selection of excitation points.

## Figures and Tables

**Figure 1 sensors-24-07302-f001:**
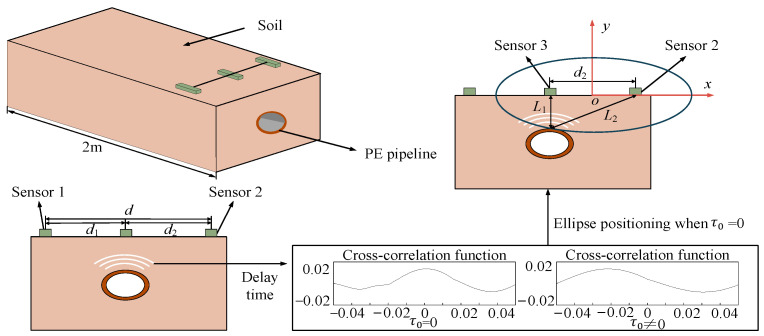
Elliptic legal bit pipeline model.

**Figure 2 sensors-24-07302-f002:**
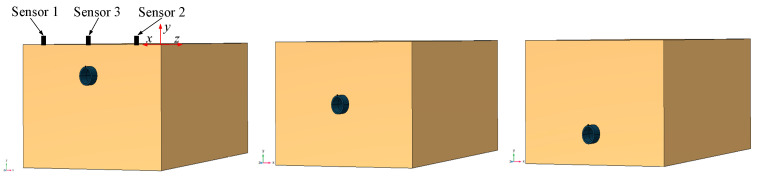
COMSOL to establish a pipeline model.

**Figure 3 sensors-24-07302-f003:**
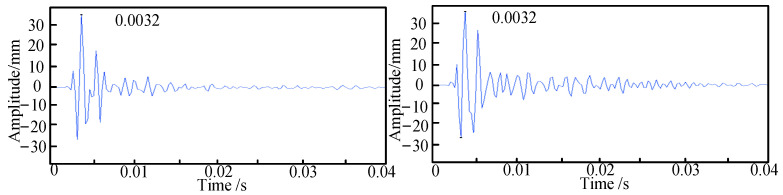
Sensors 1 and 2 collect signals.

**Figure 4 sensors-24-07302-f004:**
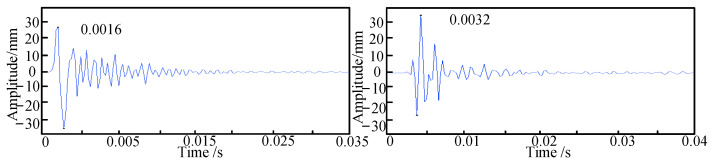
Sensors 2 and 3 collect signals.

**Figure 5 sensors-24-07302-f005:**
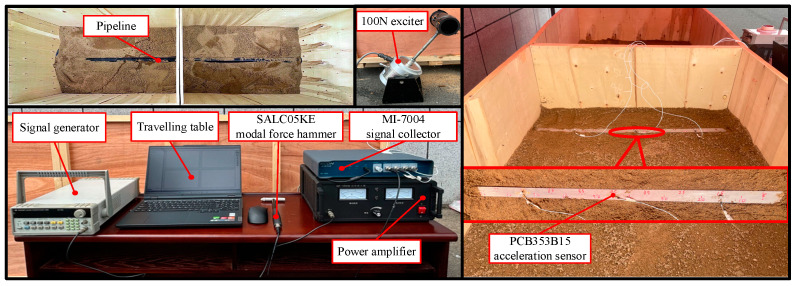
Arrangement of experimental device.

**Figure 6 sensors-24-07302-f006:**
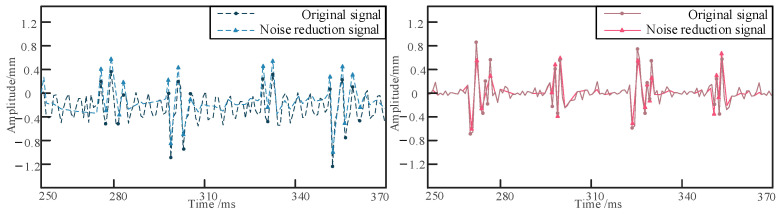
The original signal and noise reduction signal of signal 1 and signal 3.

**Figure 7 sensors-24-07302-f007:**
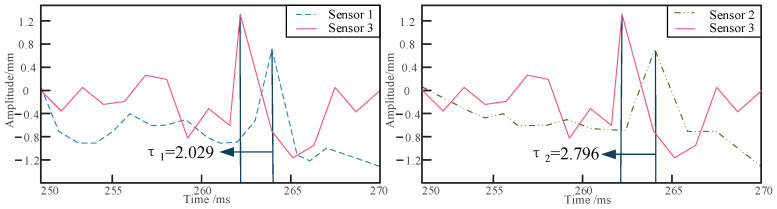
The delay time of signal acquisition at a depth of 10 cm.

**Figure 8 sensors-24-07302-f008:**
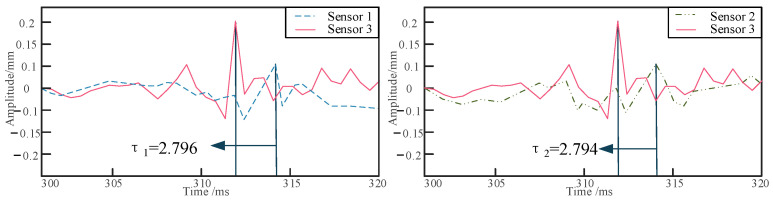
The delay time of signal acquisition at a depth of 20 cm.

**Figure 9 sensors-24-07302-f009:**
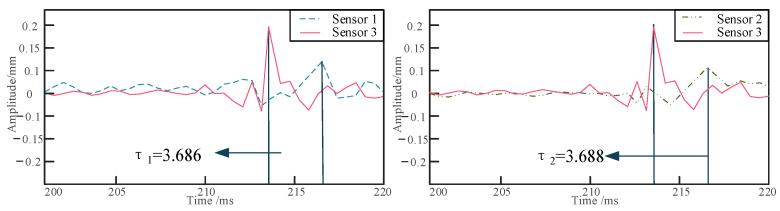
The delay time of signal acquisition at a depth of 30 cm.

**Table 1 sensors-24-07302-t001:** Material parameters of pipe and soil.

Parameter		Pipeline	Soil
Density (kg/cm^3^)	*ρ*	950	2000
Young modulus (GPa)	*E* _1_	1.08	0.083
	*E* _2_	1.08	0.083
Shear elasticity (GPa)	*G* _12_	0.38	-
Poisson’s ratio	*υ*	0.418	0.3

**Table 2 sensors-24-07302-t002:** Specific simulation scheme.

Arrangement Mode	Group	Sensor Position Coordinate		Excitation Point Position (cm)
		Sensor 1 (cm)	Sensor 2 (cm)	
The depth of the pipeline is 10 cm	First group	(−50, 0, −100)	(50, 0, −100)	(0, 0, −10)
The depth of the pipeline is 20 cm	Second group	(−50, 0, −100)	(50, 0, −100)	(0, 0, −20)
The depth of the pipeline is 30 cm	Third group	(−50, 0, −100)	(50, 0, −100)	(0, 0, −30)
The relative positions of sensors 2 and 3 are 10 cm	First group	(−30, 0, −100)	(30, 0, −100)	(0, 0, −10)
The relative positions of sensors 2 and 3 are 20 cm	Second group	(−40, 0, −100)	(40, 0, −100)	(0, 0, −10)
The relative positions of sensors 2 and 3 are 30 cm	Third group	(−50, 0, −100)	(50, 0, −100)	(0, 0, −10)
The sensor is 0.5 m away from the excitation point	First group	(−50, 0, −50)	(50, 0, −50)	(0, 0, −10)
The sensor is 1 m away from the excitation point	Second group	(−50, 0, −100)	(50, 0, −100)	(0, 0, −10)
The sensor is 1.5 m away from the excitation point	Third group	(−50, 0, −150)	(50, 0, −150)	(0, 0, −10)

**Table 3 sensors-24-07302-t003:** Simulation results for different pipeline burial depths.

Buried Depth (*H*)/cm	*τ* _0_	*y*(*K*)/cm	Error (|*H* − *K*|/*H*)
10	0.00160	10.24	2.4%
20	0.00165	20.64	3.2%
30	0.00179	31.32	4.4%

**Table 4 sensors-24-07302-t004:** Simulation results for different sensor locations.

Sensor Location (*H*)/cm	*τ* _0_	*y*(*K*)/cm	Error (|*H* − *K*|/*H*)
10	0.00165	10.32	3.2%
20	0.00164	10.28	2.8%
30	0.00160	10.24	2.4%

**Table 5 sensors-24-07302-t005:** Simulation results for different excitation point positions.

Excitation Relative Distance (*H*)/m	*τ* _0_	*y*(*K*)/cm	Error (|*H* − *K*|/*H*)
0.5	0.00165	10.30	3.0%
1	0.00160	10.24	2.4%
1.5	0.00165	10.32	3.2%

**Table 6 sensors-24-07302-t006:** Change in buried depth of pipeline.

Buried Depth (*H*)/cm	*y*_1_(*K*_1_)/cm	Error 1 (|*H* − *K*_1_|/*H*)	*y*_2_(*K*_2_)/cm	Error 2 (|*H* − *K*_2_|/*H*)
10	10.45	4.5%	10.48	4.8%
20	21.34	6.7%	21.32	6.6%
30	32.46	8.2%	32.49	8.3%

**Table 7 sensors-24-07302-t007:** Sensor location changes.

Sensor Location (*H*)/cm	*τ* _1_	*y*_1_(*K*_1_)/cm	Error 1 (|*H* − *K*_1_|/*H*)	*τ* _2_	*y*_2_(*K*_2_)/cm	Error 2 (|*H* − *K*_2_|/*H*)
10	1.234	10.87	8.7%	1.230	10.84	8.4%
20	1.599	10.62	6.2%	1.597	10.59	5.9%
30	2.029	10.44	4.4%	2.030	10.46	4.6%

**Table 8 sensors-24-07302-t008:** Sensor distance excitation position change.

**Excitation Relative Distance (*H*)/m**	** *τ* _1_ **	***y*_1_(*K*_1_)/cm**	**Error 1 (|*H*** − ***K*_1_|/*H*)**	** *τ* _2_ **	***y*_2_(*K*_2_)/cm**	**Error 2 (|*H*** − ***K*_2_|/*H*)**
0.5	2.047	10.72	7.2%	2.048	10.73	7.3%
1	2.035	10.53	5.3%	2.036	10.55	5.5%
1.5	2.045	10.69	6.9%	2.046	10.7	7.0%

**Table 9 sensors-24-07302-t009:** Excitation frequency change.

Excitation Frequency (*H*)/Hz	*τ* _1_	*y*_1_(*K*_1_)/cm	Error 1 (|*H* − *K*_1_|/*H*)	*τ* _2_	*y*_2_(*K*_2_)/cm	Error 2 (|*H* − *K*_2_|/*H*)
110	2.050	10.78	7.8%	2.051	10.79	7.9%
700	2.038	10.58	5.8%	2.039	10.61	6.1%
950	2.056	10.86	8.6%	2.054	10.83	8.3%

## Data Availability

Data will be provided upon request.

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
