# Peer review of "Location Method of Buried Polyethylene Gas Pipeline Based on Acoustic Signal Ellipse Method"

_sensors, 2024, doi:10.3390/s24227302_

Round 1
Reviewer 1 Report
Comments and Suggestions for Authors
In this paper a method is proposed and applied to detect the location of a buried pipeline using acoustic signals that are received by 3 sensors above the possible location. From the correlation in time between the received signals the location is estimated. Initial experiments indicate that the method may yield reliable estimates.
The experimental set-up and results are adequately described and analyzed, respectively. Before mentioning minor inaccuracies, typos, errors, I list my main objections:
- The theory and its presentation evoke serious questions, since they are inaccurately and incomprehensibly described. The authors analyse the signals via wavelet analysis and present the required formulae in great detail, but do not show at all that this is needed. In fact they simply have to calculate the correlation functions between signals, which can be done in a straightforward way. The whole machinery in equations (1) - (10) seems to be superfluous and is, on top of that, incomprehensible. The authors probably quote formulae from a manual or textbook, but do this is in a sloppy way, and without mentioning their source.
- In Equation (12) all of a sudden a fit-function is introduced, without any discussion. Is this choice the cause of the inaccuracies in the experimental estimates, or do these stem from other (e.g. experimental) sources.
- The series of equations (20) - (24) is so trivial that it could be compressed.
- In the experiment one pipeline is used (line 288). However, in the text (lines 309, 324, 332,..) the authors confusingly mention 2 pipelines.
- The discussion contains much repetition. Cf. lines 367-371 with 372 - 376; and 398 - 402 with 403 - 407.
Minor remarks:
- l.134: proposed -> proposes
- l.156: a 'tau' is used, but later on this is a 'T' (e.g. in Fig. 1)
- l.166: superfluous comma
- In (2) the l.h.s the 'j' has subindex 1, but in the r.h.s. not
- In (6) the sums run over p, resp. q, but the expressions do not contain any p or q....
- l.183 and 185 and 186: strange notation, with a subindex of a subindex..
- l.200: 18 must probably be 13
- l.222: p-wave velocity is not defined
The methods applied in this paper are quite straightforward, but might be useful in practice. However, the theory is presented in an highly inaccurate and incomprehensible way. The authors probably cite formulae from standard books without referring to these sources. The presentation in section 2 in its present form does not make much sense to the reader and is is hardly informative.
Comments on the Quality of English LanguageThe english is acceptable, although improvements are possible.
Reviewer 2 Report
Comments and Suggestions for Authors
Pipeline damage localization is an important research direction in structural health monitoring. This manuscript conducts relevant research on buried pipelines, which is a very interesting piece of work. The manuscript can be considered for publication, but some revisions are needed. The suggested modifications are as follows:
1. The numerical simulation section is too simplistic and needs to include more computational details.
2. The current model has sensor 2 positioned directly above the pipeline. Is this method applicable when sensor 2 is placed at other positions between sensors 1 and 3?
3. Does the method presented in this paper require prior knowledge of the buried pipeline's location?
4. What is the maximum distance that can be monitored using the method described in the manuscript?
5. The sensors need to detect the acoustic waves generated by pipeline leaks. In practical applications, how is the coupling effect between the sensors and the soil?
6. It is recommended to add the time delay results of the numerical simulation waveforms to correspond with the experimental section of the paper.
7. The waveforms in the manuscript do not account for interference from noise and other factors. What are the reasons for the errors generated in the numerical simulation presented in Table 3?
8. The waveforms collected from the experiments in the manuscript should be presented.
9. It is recommended to include some literature on localization in the manuscript to enhance the depth and breadth of the research, such as:
1. Huang X, Xu R, Yu W, et al. Impact Localization in Complex Cylindrical Shell Structures Based on the Time-Reversal Virtual Focusing Triangulation Method[J]. Sensors, 2024, 24(16): 5185.
2. Chen B, Hei C, Luo M, et al. Pipeline two-dimensional impact location determination using time of arrival with instant phase (TOAIP) with piezoceramic transducer array[J]. Smart Materials and Structures, 2018, 27(10): 105003.
3. Liu J, Hei C, Luo M, et al. A study on impact force detection method based on piezoelectric sensing[J]. Sensors, 2022, 22(14): 5167.
4. WITOS F, OLSZEWSKA A, OPILSKI Z.. LOCATION OF LEAKAGE IN METAL PIPELINES BY THE ACOUSTIC EMISSION METHOD USING MODELED AND NATURAL SOURCES[J]. ACOUSTICS, ACOUSTOELECTRONICS AND ELECTRICAL ENGINEERING, 404.
Comments on the Quality of English LanguagePipeline damage localization is an important research direction in structural health monitoring. This manuscript conducts relevant research on buried pipelines, which is a very interesting piece of work. The manuscript can be considered for publication, but some revisions are needed. The suggested modifications are as follows:
1. The numerical simulation section is too simplistic and needs to include more computational details.
2. The current model has sensor 2 positioned directly above the pipeline. Is this method applicable when sensor 2 is placed at other positions between sensors 1 and 3?
3. Does the method presented in this paper require prior knowledge of the buried pipeline's location?
4. What is the maximum distance that can be monitored using the method described in the manuscript?
5. The sensors need to detect the acoustic waves generated by pipeline leaks. In practical applications, how is the coupling effect between the sensors and the soil?
6. It is recommended to add the time delay results of the numerical simulation waveforms to correspond with the experimental section of the paper.
7. The waveforms in the manuscript do not account for interference from noise and other factors. What are the reasons for the errors generated in the numerical simulation presented in Table 3?
8. The waveforms collected from the experiments in the manuscript should be presented.
9. It is recommended to include some literature on localization in the manuscript to enhance the depth and breadth of the research, such as:
1. Huang X, Xu R, Yu W, et al. Impact Localization in Complex Cylindrical Shell Structures Based on the Time-Reversal Virtual Focusing Triangulation Method[J]. Sensors, 2024, 24(16): 5185.
2. Chen B, Hei C, Luo M, et al. Pipeline two-dimensional impact location determination using time of arrival with instant phase (TOAIP) with piezoceramic transducer array[J]. Smart Materials and Structures, 2018, 27(10): 105003.
3. Liu J, Hei C, Luo M, et al. A study on impact force detection method based on piezoelectric sensing[J]. Sensors, 2022, 22(14): 5167.
4. WITOS F, OLSZEWSKA A, OPILSKI Z.. LOCATION OF LEAKAGE IN METAL PIPELINES BY THE ACOUSTIC EMISSION METHOD USING MODELED AND NATURAL SOURCES[J]. ACOUSTICS, ACOUSTOELECTRONICS AND ELECTRICAL ENGINEERING, 404.
Round 2
Reviewer 1 Report
Comments and Suggestions for Authors
The authors improved the original submission by taking into account the suggestions for improvement. E.g., the Discussion has considerably been shortened, so that repetition is no longer the case.
However, the paper has still one very weak aspect, which is so serious that it prohibits publication in this form, in my opinion. It concerns the theory presented in equations 1 - 8.
In the revised paper the authors do not explain why they pay so much attention to the wavelet theory. In 4.1 they reveal that it is necessary to denoise the measured signals. This motivation should be mentioned earlier.
More seriously, they probably copy formulae 1 - 8 from some source, that is not referenced at all, but do this erroneously and without any explanation. It seems as if they do not understand this theory and in practice just applied a software package. If this is true it would be fair to skip equations 1 - 8 and replace it by the remark that "the experimental signals were denoised using package '.....' ," and refering to the manual of that package.
To underpin my skepsis, I mention some flaws in the formulae:
- In the r.h.s. of Eq. 4a,b a 'j' is mentioned , which has no meaning. Maybe an 'i' meant.
- In the same Eq. 4a,b the small 'c' and the capital 'C' are mixed up in a strange way.
-In Eq. 7 the index 'j' on the r.h.s. has no meaning. Has the expression to be summed over j or so? This 'j' is absent in the l.h.s.
Since Eqns 1 - 8 do not contribute to the essence of the paper, I suggest to leave them out, as stated above.
Other remarks:
- The sentence: 'The two signals 𝑥̃1(𝑡) and 𝑥̃2(𝑡) are respectively replaced by x1(x,t) and x2(x,t),' in line 200 is unclear. Furthermore the subindices should be corrected
- The sentence in line 209 is not correct english.
Comments on the Quality of English Language
No comments
Reviewer 2 Report
Comments and Suggestions for Authors
The author has made appropriate revisions to the manuscript, resulting in significant improvements to the content of the paper. It is recommended for publication.
Author Response
Thank you for your review and valuable comments!